# (Mis)estimation of the modal number of desired sexual partners

**Arthur G. Shapiro** *, **Rubie M. Peters, Anthony H. Ahrens**

American University, Washington, DC, United States of America

* arthur.shapiro@american.edu

## Abstract

Misperceptions of the social world can lead to actions and social policy that are detrimental to an individual's or group's well-being. Here we investigate whether misperceptions arise when participants make predictions of the *modal* number of ideal future sexual partners reported by heterosexual cohorts (younger cohort: 18–23 years; older cohort: 24–29 years). For both men and women and in both cohorts, the modal number of reported partners equaled 1.0, but men's averages were higher than women's averages due to a subgroup of men who reported desiring large numbers of partners (that is, the distributions had the same shape, but men's distributions had a longer tail). Study 1: When asked to estimate the mode directly, participants performed poorly and, in some conditions, dramatically so (e.g., 40% of younger men reported wanting one sexual partner, but 0% of younger men predicted 1 to be the most frequent response). Study 2: When asked to estimate the shape of the whole distribution, participants underestimated the number of respondents who would desire the mode and thus replicated patterns in the literature for misestimations of skewed distributions. Study 3: When provided information about others' actual modal desired number of partners, the number of male participants who reported desiring one sexual partner increased, suggesting that misperceptions of social norms may influence preferences. We discuss how the mean and mode can lead to two accurate but different interpretations of the data (mean: men report desiring more sexual partners than women; mode: the most frequent response reported by both men and women is 1.0). Discrepancies of this sort can lead to mischaracterizations that may not be uncommon in the research literature. These discrepancies cannot be differentiated by significance tests that seek to find differences in the mean but can be resolved with attention to other methods of analyses.

## Introduction

An "illusion" occurs when perception conflicts with a measurable aspect of the physical or social environment, or when perception is unable to integrate multiple sources of conflicting information into a stable expected interpretation [1–3]. Illusions are of longstanding interest because they illustrate how our brains construct reality, and they have proven to be fundamental for understanding the processes underlying perception [4], cognition [5], economics [6],

**Data Availability Statement:** We have updated the location of the datasets. The data are now located on OSF repository: https://osf.io/w2hz8/.

**Funding:** The author(s) received no specific funding for this work.

**Competing interests:** No authors have competing interests

and social perception [7], and the implications of these processes. The organization Gapminder, for instance, has extensively documented that people on average are much worse than chance at identifying positive trends in world public education, world wealth, frequency of diseases, effectiveness of immunization treatments, life expectancy, childhood mortality, and other public health data that show improvement over time [8].

One might expect that such misperceptions (or "estimation illusions") are the rule, particularly since misperceptions are often the basis of public demonstrations [9], and misperceptions and distortions often illustrate psychological phenomena (see [10]). However, historically, "Wisdom of the Crowd" theories suggest that collective estimates can be more accurate than those of an individual. Galton [11] famously tallied estimates of the weight of an ox made by 787 people at a Plymouth County Meat and Poultry exhibition and found that the central tendency of the estimates (1207 pounds) was within 1 percent of the actual weight of the ox (1198 pounds lb.), even though the set of estimates contained a considerable number of over- and underestimations. While there have been numerous anecdotes that show that collective estimates of ordinary observers are more accurate than experts' estimates [12], and there is evidence that wisdom-of-crowd principles can improve on individual estimates of population means [13], there are also reasons to doubt the ability of pooled analyses to outdo experts [14] and to doubt that wisdom-of-crowd theories will save from us from systematic errors (see [10], pp. 152–158).

Part of the reason for the discrepancy between "estimation illusions" and wisdom-of-the-crowd accuracy is that the ability to make predictions depends on the situation. For instance, on average, people make impressively accurate estimates for variables with symmetric population distributions [15] but make systematic errors when estimating skewed distributions [15, 16]. Nisbett and Kunda [15] found that participants' estimates of the (fairly normally distributed) frequency with which people go to parties were relatively accurate. In contrast, estimations of marijuana usage were quite inaccurate. This latter distribution was heavily skewed, and participants underestimated the frequency of non-use. Several theoretical approaches seem to be able to account for inaccurate estimates for skewed distributions–for instance, a bias towards normality for observers outside the standard range [15], social sample theory [17], and effects of moderately sized salient outliers [18, 19]. Even representation or availability heuristics [20] would predict misestimation of skewed distributions if items at the tail end of the distribution were more salient than other elements.

While observers, on average, often make accurate estimates of the means of symmetric distributions and inaccurate estimates of the means of skewed distributions, it is not clear what the mis-estimation implies for the estimation of different summary statistics (e.g., mean, median, mode, or location statistics). In this paper, we concentrate on the estimation of the mode: the mode is an often-ignored statistic because the sampling properties are not as convenient as the mean, and if the sample size is small, then the mode's value can be changed by a few spurious numbers. Nonetheless, as observed by Chacón [21], for skewed distributions the mode can capture more of the data than the median and mean (in Chacón's example, the mode was closest to the shortest interval containing 50 percent of the distribution data). Questions about how to characterize distributions are particularly important because many distributions in the social sciences are skewed [22, 23], and reliance on the mean to describe skewed distributions can lead to misinterpretations [24–26].

## Social relevance of misestimation

Galton [11] began his article about estimating the weight of an ox with, "In these democratic days, any investigation into the trustworthiness and peculiarities of popular judgements is of

interest." The same sentiment still holds true; misperceptions of the social world like those reported by Gapminder are important to understand because they can lead to social policy and actions that are detrimental to an individual's or group's well-being, while identification of such misperceptions can lead people to adjust their behavior in a beneficial manner. For instance, pluralistic ignorance can lead people to act based on misperceptions of the behavior of others [27, 28]. As one example, college students consistently overestimate how likely their peers are to engage in binge drinking [29], and this leads people who make overestimations to engage in more binge drinking. As an instance of social norms interventions [30, 31], some evidence suggests that students presented with accurate information about the frequency of binge drinking are less likely to engage in harmful drinking behaviors [32, 33], for a failure to replicate).

Another example of the consequences of the misperception of social statistics can be found in estimation of sexual behavior. Popular sources seem to suggest that people have more partners nowadays than was once the case (e.g., [34]), but research contradicts the popular accounts (e.g., [35–37]). This mismatch of popular accounts and research findings suggests that there is a misperception of the amount of other people's sexual activity (e.g., [28]). Indeed, people overestimate how comfortable people are with casual sexual behavior (e.g., [27, 28, 38, 39]), and how much pornography people consume [40]. If people become aware of this misperception, the awareness might matter for behavior. A recent meta-analysis found a strong association between descriptive norms of peer sexual activity and adolescent sexual activity [41]. Given this association, some authors have called for social norms interventions in this area [42].

## The present study

The problem of using a summary statistic to describe the central tendency of skewed distributions is illustrated in the distribution for survey responses to the following question: "How many sexual partners would you ideally like to have over the next 30 years?" Buss and Schmitt [43] originally reported that on *average*, males reported desiring more sexual partners than females (thus supporting the idea that males desire more short-term partners than females). However, Pedersen et al. [44] showed that the distribution of the number of sexual partners desired was highly skewed; hence, while the *mean* for male responses was higher than the mean for females, the *median* was similar (thus supporting the idea that humans, regardless of gender, desire more long-term pair-bonding), and the *mode* for both females' and males' distributions was 1. The data therefore indicate that the distributions of men's and women's desired numbers of partners are similar, except that there is a larger proportion of extreme values in the male distribution. Similarities between women's and men's responses to this and similar survey questions are often noted in the vast literature concerning measurements of sexual behavior (see, for instance, [45, 46]), however, if observer responses are described only by the mean, these similarities will not be reported.

In the studies presented in this paper, we ask participants to estimate the most frequent response (i.e., the modal response) to the question "How many sexual partners would you ideally like to have over the next 30 years?" Our goal is both to replicate Pedersen et al.'s [44] finding, and then to explore whether people misestimate the proportion of women and men who desire only one sexual partner over the next 30 years. In Pedersen et al.'s sample, the male distribution has a greater skew than the female distribution. We therefore expect that there would be a misperception of the others' preferences, and that this misperception would be greater for estimates of men's reported preference than for women's preference. We also address whether providing information about what is known about the underlying distribution prior to survey

completion can affect people's reports of desired number of partners. The last experiment mirrors studies that show that correcting misperception of drinking norms may reduce college students' alcohol use [30, 47].

In our data, we show that different measures of centrality produce different interpretations: the mean by itself leads to one conclusion (that men desire more partners than women), whereas the mode by itself leads to another conclusion (that men and women report desiring a similar number of partners; the groups differ primarily because a small cohort of men report desiring a high number of partners). These two interpretations cannot be differentiated by significance testing of the mean–the standard method of analysis for studies in Psychology. We suggest that the conflict between these interpretations could arise for any study that samples from a skewed population distribution. These conflicts can be resolved by considering multiple measures of centrality [21] and other statistical procedures [26].

## Study 1: Documentation of misperception of the mode

Study 1 investigates whether participants can predict the most frequent number of sexual partners desired by their peers. The study replicates Pedersen et al.'s [44] measurements and then asks participants to make a prediction about the results of the study: i.e., what will be the most frequent response given by men and by women? Given the expected skew of the distributions, we hypothesized that relatively few participants would predict that their peers would most frequently respond that they desire only one sexual partner.

### Method

**Participants.** Participants were recruited through Amazon's Mechanical Turk (MTurk; [48]). Participants were initially given 15¢, but compensation was later increased to 30¢ to increase the response rate. Of the 818 MTurkers who selected the link to the Qualtrics survey, 351 subjects qualified via the filter questions (i.e., U.S.-born males and females between 18 and 29 years old). Of these, 84 participants either did not report their sexual orientation ($n = 25$) or reported being LGBTQ ($n = 59$). We restricted our data analysis to the 267 heterosexual participants (141 males and 126 females) with the aim of collecting a sufficiently large sample of a relatively homogenous population. The average age for heterosexual participants was 24.52 ($SD = 3.07$). Sixty-six participants were paid 15¢, and 201 were paid 30¢. Payment types were combined for analyses. One participant did not provide predictions for others and so was not included in analyses that involve predictions for others. Two participants predicted zero as the most frequently desired number of partners; these two data points are not depicted in plots of the data but are included in analyses.

Studies one and two were approved by the American University Institutional Review Board (approval number #15000). All participants gave consent by clicking on an "I agree" button on the Qualtrics system after they read the conditions of the study. The data collection for Study 1 took place between January 30 and March 3, 2015.

**Measures.** Two questions were adapted from previous research [43, 44]. Participants were first asked about their ideal number of sexual partners–"How many sexual partners would you 'ideally' like to have over the next 30 years?"–and then were asked to predict their peers' desired number of sexual partners. The peer prediction questions were asked for men and women of participants' age cohort (younger: 18–23; or older: 24–29). For example, both male and female 18- to 23-year-old participants were asked, "What would be the most frequent response by heterosexual female MTurk users between the ages of 18 and 23 from the United States?" and "What would be the most frequent response by heterosexual male MTurk users between the ages of 18 and 23 from the United States?"

**Procedure.**   Participants gave informed consent and then answered a set of filter questions to limit the sample to heterosexual U.S. citizens that identify as male or female and who are in the appropriate age range for the study. Participants who met the filter criteria and continued with the study were asked about their own ideal desired number of sexual partners, and then their perception of same-age peers' desires (the order of these two questions was randomized). Participants were then asked questions concerning their technology usage, sexual orientation, number of previous sexual partners, current relationship status, race, and political affiliation. Participants were thanked and given a randomly generated survey code to redeem the reward for completing the survey in its entirety.

## Results

Fig 1 shows the distribution of responses to questions about participants' desired number of partners and their predictions of others' responses. The data are presented as four separate graphs, one for each age and gender group (panel A, Younger Females; panel B, Younger Males; panel C, Older Females; panel D, Older Males). Each plot shows the frequency distributions of each response (i.e., the percentage choosing a particular number of partners). We have binned the values on the x-axis to follow an exponential scale (that is, 1, 2–3, 4–7, 8–15,16–31, 32–63, 64–127, >128) since the values of desired sexual partners are skewed in a typical Weber-Fechner pattern [49], (see also [50], for a discussion about proportion and number scaling).

The reported desired number of sexual partners is shown as filled black circles. As can be seen in the plots, the most frequent response (the mode) for all four groups is 1; the percentage of each group reporting 1 is as follows: younger women: 51.67%; younger men: 40.00%; older women: 60.61%; older men: 39.60%. The means, medians, and modes for desired number of partners for each age and gender group are presented in Table 1. The mean for younger males was 517.70 (SE = 215.0, 95% CI [93.5, 942.0]), and the mean for older males, was 108.90 (SE = 136.0, 95% CI [-158.0, 376]). The mean for younger females was 3.70 (SE = 176.0, 95% CI [-342.7, 350.0]) and the mean for older females was 2.50 (SE = 168, 95% CI [-327.7, 333.0]). The medians were 3 for males and 1 for females. Using a 2 (age) x 2 (gender) ANOVA, there was no evidence of an effect of gender [$F_{(1, 263)} = 3.107$, p = .079], or of age [$F_{(1, 263)} = 1.357$, p = .245]. There were two extreme responses (a younger male reported wanting 20000 partners and an older man reported wanting 10000 partners, whereas no one else wanted more than 300 partners. Given these extreme responses, we ran a Kruskal-Wallis test that showed that the mean rank of desired partners was affected by gender, $\chi^2(1) = 10.21$, p < .002, but found no evidence of an effect of age, $\chi^2(1) = 1.77$, p = .183.

The predictions of the most frequent (modal) response are shown in Fig 1 as purple squares (predictions by same-aged women) and as green diamonds (by same-aged men). All groups had the highest number of respondents predicting between 2–3 (younger women) and 4–7 (older women) and between 8–15 for younger and older men. There was no evidence of differences in the mean rank of predictions made by men versus women about the number of partners desired by women ($\chi^2(1) = 1.05$, $p = .306$) or by men ($\chi^2(1) = 3.756$, $p = .053$). But 230 participants did report that they thought men would desire more partners than women, whereas only 10 participants thought that women would desire more partners than men.

Most striking, though, was that only 9.74% of the respondents correctly predicted that the modal response for same-age, same-gender subjects would be 1. To capture this misperception, we use a statistic $D_{mismatch}$, which equals the percentage of participants who reported desiring exactly 1 partner minus the percentage of those of that age/gender group that predict

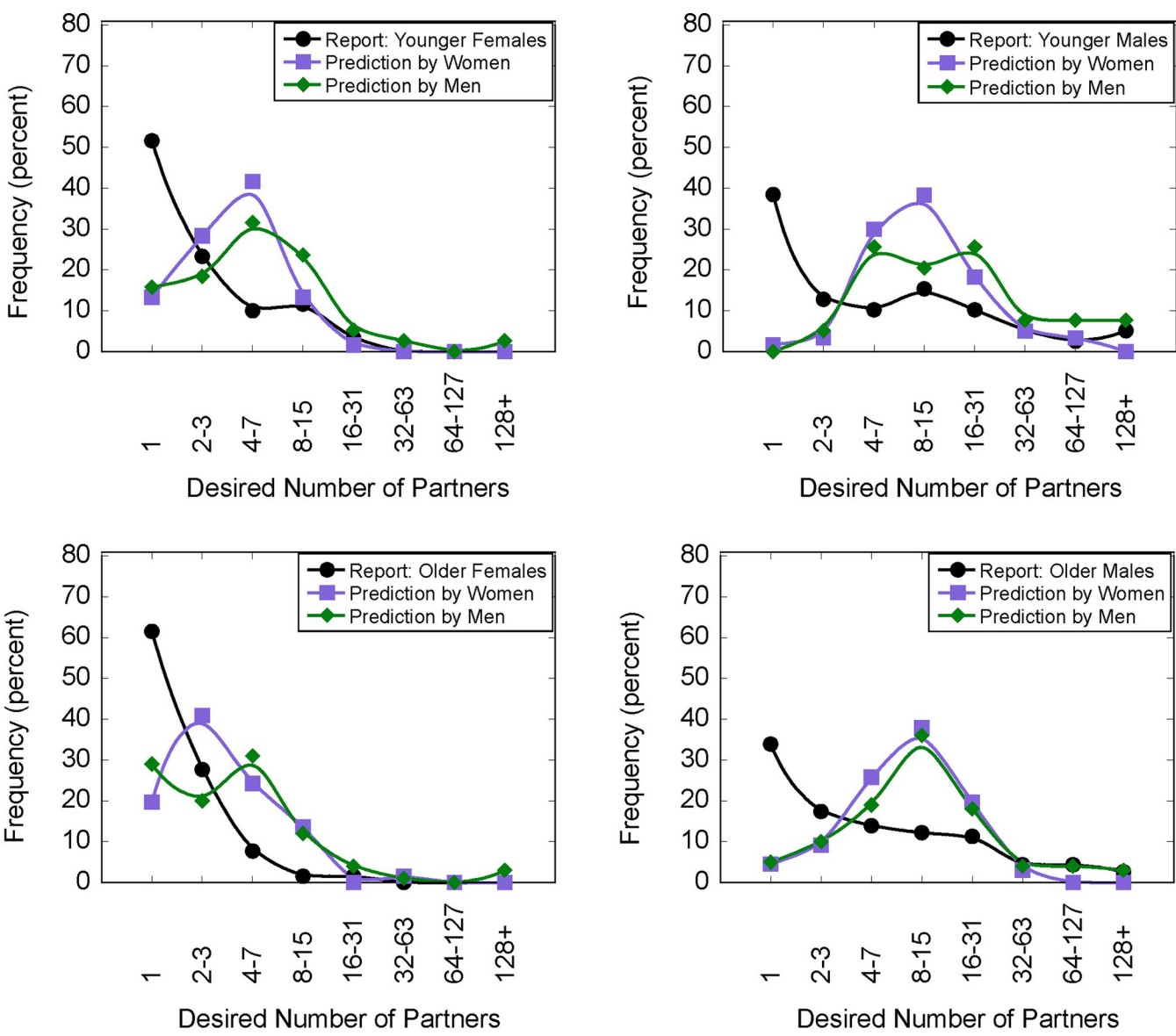

**Fig 1. Results of Study 1.** Filled black circles indicate the frequency of reported number of sexual partners ideally wanted over the next 30 years reported by A) Younger Women (ages 18–23), B) Younger Men (ages 18–23), C) Older Women (ages 24–29), and D) Older Men (24–29). The same participants also answered the question "What would be the most frequent response given by [Men/Women]" in the same age group. The purple squares indicate frequency of the predictions by same-aged female participants; green diamonds by same-aged male participants.

1 being the most frequent response. We used judgments about men and women, by both men and women, not just the same-age, same-gender judgments. $D_{mismatch}$ values for predictions about different age/gender groups are as follows: Younger Women, 36.5%; Younger Men, 39.0%; Older Women, 35.3%; Older Men, 34.8%. These values are approximately the same for all four groups. Lastly, we note that those who predicted that others would desire more partners also desired more partners for themselves $r_S(265) = 0.358$, p < .001), 95% CI [0.245, 0.462]. This is similar to previous research that perception of the risky sexual behavior of peers correlated with subjects' own past sexual behavior [39].

**Table 1. Mean, median, and number of participants reporting one partner for studies 1, 2, and 3.**

| | Age group | n | Mean (SD) | Median | Number reporting 1 (%) | n | Mean (SD) | Median | Number reporting 1 (%) |
|---|---|---|---|---|---|---|---|---|---|
| Study 1 | | | Females | | | | Males | | |
| | 18–23 | 60 | 3.70 (4.92) | 1 | 31 (51.67%) | 40 | 517.70 (3,159.80) | 3 | 16 (40.00%) |
| | 24–29 | 66 | 2.52 (4.32) | 1 | 40 (60.61%) | 101 | 108.90 (994.25) | 3 | 40 (39.60%) |
| Study 2 | | | Females | | | | Males | | |
| | 18–23 | 70 | 3.53 (7.78) | 1 | 42 (60.00%) | 84 | 11.83 (44.72) | 3 | 34 (40.48%) |
| | 24–29 | 85 | 2.01 (2.53) | 1 | 65 (76.47%) | 90 | 27.23 (118.90) | 3 | 41 (45.56%) |
| Study 3 | | | Females | | | | Males | | |
| | | | | | Average Condition | | | | |
| | 18–23 | 99 | 3.64 (5.38) | 1 | 52 (52.53%) | 100 | 31.50 (110.07) | 5 | 35 (35.00%) |
| | 24–29 | 98 | 3.64 (15.21) | 1 | 67 (68.37%) | 99 | 27.96 (108.13) | 3 | 39 (39.40%) |
| | | | | | Modal Condition | | | | |
| | 18–23 | 100 | 2.98 (5.91) | 1 | 62 (62.00%) | 100 | 23.06 (111.98) | 2 | 46 (46.00%) |
| | 24–29 | 102 | 2.48 (4.42) | 1 | 69 (67.65%) | 100 | 7.71 (19.80) | 1 | 55 (55.00%) |
| | | | | | Control Condition | | | | |
| | 18–23 | 100 | 3.74 (10.86) | 1 | 62 (62.00%) | 99 | 12.22 (50.59) | 3 | 37 (37.37%) |
| | 24–29 | 100 | 3.73 (10.22) | 1 | 67 (67.00%) | 99 | 14.96 (33.84) | 3 | 43 (43.43%) |

## Discussion

Study 1 replicated the Pedersen et al. [44] distributions on an MTurk population. The male and female distributions were positively skewed, with a mode of 1. Participants' predictions of the most frequent desired number of sexual partners in their age group did not match the reported values of their peers; indeed, less than 10% of participants accurately predicted that the most common desired number of sexual partners for same-age, same-gender peers would be 1. There was no evidence that men's and women's predictions differed from each other. Overall, participants predicted that men's most frequent response would be higher than women's most frequent response.

## Study 2: Estimating the shape of the distribution

Study 1 asked participants to estimate the "most frequent number" (the mode) of sexual partners reported as ideal by participants in their peer group. Since estimating the mode is an unusual procedure, it was unclear to us whether the participants understood that they were estimating the mode of the distribution or whether they thought they were estimating some other type of statistic. To better understand the participants' conception of the underlying distribution, we repeated Study 1 but asked them to estimate the values for the entire frequency distribution, thereby following a procedure similar to previous research (e.g., [16, 17]).

## Method

### Participants

MTurkers were paid 30¢ to take part in the study; their initial responses were filtered with the same questions as in Study 1 (i.e., participants were U.S.-born males and females between 18 and 29 years old). Of the 1,845 MTurkers who initiated the survey, the filter questions identified only 393 in the appropriate age range for this study. As with Study 1, we restricted our data analysis to 330 heterosexual participants (175 males and 155 females) and removed 63 non-heterosexual participants. The average age for the heterosexual participants was 23.98 (SD = 3.1). One participant did not complete the "desired number of sexual partners" question,

so their results were not included in analyses involving that variable. Two participants (one younger female and one older male) reported desiring 0 partners; their results are not presented in the figure presenting Study 2 results but are included in the analyses. The data collection took place between August 25 and September 1, 2015.

**Measures.**   Participants were told that others on MTurk had been asked: "How many sexual partners would you ideally like to have over the next 30 years?" They were then asked to indicate the percentage of responses from a given group that they thought fell within a particular range. That is, what percentage of responders answered one sexual partner, 2–3 sexual partners, 4–7, and so on. The categories were the same as those used in Study 1, plus the option of zero partners. Participants were told that the sum of their responses should add to 100 percent. Qualtrics required this condition to be met before subjects could move to the next screen. Participants were asked to estimate the percentages for male and female heterosexual MTurkers of their age group; the order of estimation of male preferences and female preferences was randomized.

**Procedure.**   The procedure was the same as in Study 1, except that we dropped the questions about technology, and kept the question about subjects' own desired number of partners.

## Results

As with Study 1 and Pedersen et al.'s study [44], the distribution of males' and females' desired number of partners (filled black circles in Fig 2) is highest at 1 (the mode) and trails off with a long tail. The average desired numbers of sexual partners are as follows: younger females: 3.53 (SD = 7.78), 95% CI [1.67,5.38]; younger males: 11.83 (SD = 44.72), 95% CI [2.13, 21.54]; older females: 2.01 (SD = 2.53), 95% CI [1.47, 2.56]; and older males: 27.23 (SD = 118.90), 95% CI [2.47, 52.00]. On average, men desired more partners than did women: $F(1, 325) = 5.21$, $p = 0.023$.

The modes were 1.0 for all groups. The percentage reporting a value of 1 for each group was as follows: younger females: 60.0%; younger males: 40.48%; older females: 76.47%; older males: 45.56%; hence the frequency of reporting that one desires only 1 partner is highest for the older females and lowest for the younger males. The means, medians, and modes for desired number of partners for each age and gender group are presented in Table 1.

In each plot of Fig 2, the predicted distributions averaged across the participants are shown as purple squares (predictions by same-aged females) and green diamonds (by same-aged males). There are two central observations. First, unlike Study 1, the mode of the predictions averaged across observers was 1 for three of the conditions (Panels A, C, D). That is, for these panels the average estimate for how often others would desire exactly one partner was larger than any of the other average estimates. This was not the case for predictions about younger males (panel B), whose mode was between 4 and 7 for predictions by both males and females.

So, when participants were asked to estimate the whole distribution, the mode of these estimates was one sexual partner in three of the four estimated distributions–a result that suggests that participants' "most frequent" estimates in Study 1 are different from what participants consider to be the most frequent in terms of the whole distribution. A similar point can be made by examining which category received individuals' highest estimates, akin to the mode. For instance, if a participant thought 40% of others would desire exactly one partner, but 30% would desire 2–3, and another 30% would desire 4–7 partners, that participant's "mode" (highest estimate) would be 1. For predictions by males, 37.5%, 16.1%, 53.2%, and 25.4% had 1 as the mode for younger women, younger men, older women, and older men, respectively. For predictions by females, 37.9%, 19.5%, 46.6%, and 22.4% had 1 as the mode for those respective categories. So, while more subjects had 1 as the mode using the Study 2 procedure than the

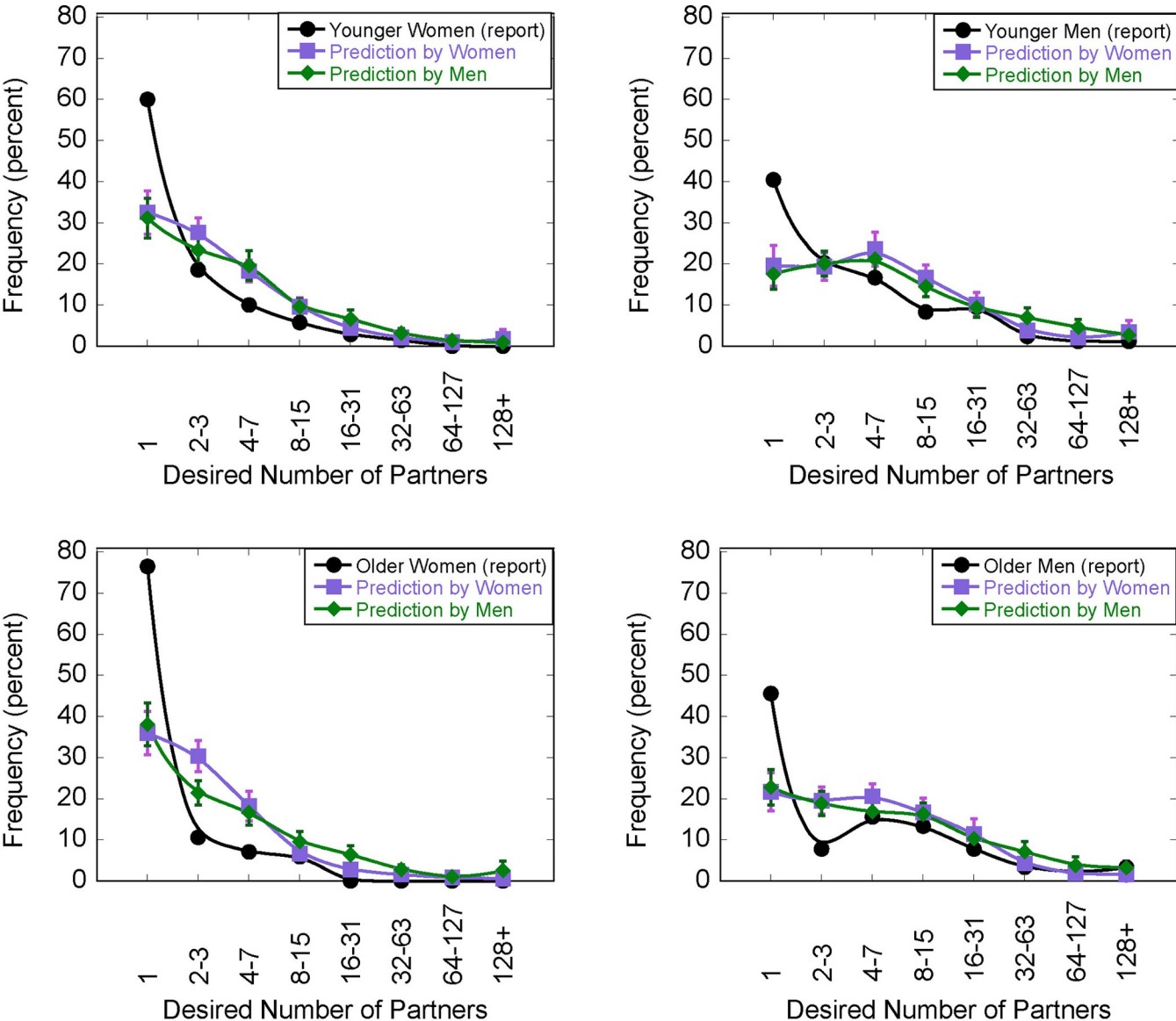

**Fig 2. Results of Study 2.** Filled black circles indicate the frequency of reported number of sexual partners ideally wanted over the next 30 years reported by A) Younger Women, B) Younger Men, C) Older Women, and D) Older Men. The same participants were also asked to indicate the percentage of responses from a given group that they thought would fall within a particular range. The purple squares indicate frequency of the predictions by same-aged female participants; green diamonds by same-aged male participants. Bars indicate +- 1 standard error.

9.8% in Study 1, on the individual analysis, most participants did not correctly identify 1 as the most frequent response.

Second, even though some of the average prediction-distributions had a mode of 1, similar to Study 1, the predictions greatly underestimated how many participants desired a single partner. Averaged across age and gender groups, participants estimated that just 26.90% of same-gender, same-age people would desire exactly 1 partner, whereas 40.48–76.47% of participants reported one partner (depending on age and gender). The discrepancy can be expressed as $D_{mismatch}$ (i.e., percent of participants who reported a value of 1 minus the participants' prediction of the percent that would desire 1 partner). The average values of $D_{mismatch}$ (calculated for all estimates, not just the estimates of same-sex subjects) are as follows: Younger Women,

28.23%; Younger Men, 22.01%; Older Women, 39.42%; Older Men, 23.28%. To compare mis-perceptions of women's versus men's reporting that they desire exactly 1 partner, we performed a paired-sample t-test, comparing each subject's misestimations for same-aged men (M = 22.69%, SD = 20.13%) vs. same-aged women (M = 34.18%, SD = 24.56%). Subjects were much less accurate in their estimates for women, $\mu$M 95% CI [9.34%, 13.54%], (t (328) = 10.72, p < .001, d = 0.591).

Lastly, as with Study 1, we examined the relationship between a participant's reported desired number of partners, and their estimates of others' desired number of partners. Participants who wanted more partners themselves gave lower estimates of the percentage of similar others that would desire one partner ($r_S$ (328) = -0.433, 95% CI [-0.519, -0.338], p < .001).

## Discussion

Participants were asked to estimate the whole distribution of participants' responses as opposed to simply giving an estimate of the mode statistic. As with Study 1, participants underestimated the proportion of participants who would give a response of 1. However, unlike Study 1, the average of estimates for three of the sample groups had a mode of 1. So, while both Studies 1 and 2 show an underestimation of the number of participants who desire a single partner, the qualitative description of the misestimation depends on whether observers are asked to estimate the most frequent response or asked to estimate the shape of the distribution.

## Study 3: Do accurate normative messages shape desired number of partners?

We have found that subjects have incorrect perceptions of the number of sexual partners others desire. For instance, in Study 1, the most frequent response given by men was that they desired only one sexual partner over the next thirty years, but less than 4% of men estimated that one would be same-aged men's most frequently desired number of partners. Presumably, then, there are numerous men and women who desire only one sexual partner but think that desiring one partner would be an atypical condition. Since providing correct information about statistical norms has been shown to influence behavioral responses [32], we tested the hypothesis that presenting information about the typicality of desiring one sexual partner would reduce the number of partners that participants will report desiring for themselves.

### Methods

**Participants.** As with the two previous studies, participants were recruited through Amazon's Mechanical Turk (MTurk) service. Of the 12,551 MTurkers who selected the link to take the Study, 1,200 passed through the filter questions; the selection was set to ensure 300 participants in each group. The final sample was 1196 subjects: 599 females, 597 males; four of the participants did not provide information necessary to complete the study. The initial participants (*n* = 943) were paid 30¢ for completion of the study; to increase the rate of response, we increased the payment to 55¢ to attract the final 253 participants. Payment types were combined for analyses.

Study 3 was approved by the IRB for the University of Mississippi on October 25th, 2016, (Protocol #17x-082*)* where Rubie Peters was at that time enrolled as a doctoral student. All participants gave consent by clicking on the Qualtrics system's "I agree" button after they read the conditions of the study. The data collection took place between November 5 and December 24, 2016.

**Measures.** Participants were randomly assigned to groups that received normative messages about (a) the mode for previous same-aged, same-gendered subjects, along with the

**Table 2. Statistics used in normative messages in Study 3.**

| Group | Mean | Mode (percent of group desiring one future sexual partner) |
| --- | --- | --- |
| Young Females | 4 | 56% |
| Older Females | 2 | 70% |
| Younger Males | 175 | 40% |
| Older Males | 70 | 42% |

percentage choosing the mode ("mode condition"); (b) the average of previous same-aged, same-gendered subjects' desired number of sexual partners ("average condition"); or (c) the fact that previous subjects had been asked their desired number of sexual partners, but they were not give information about the prior groups' statistics ("control condition"). The wording of the information provided to the groups was as follows: "We previously ran a study on MTurk in which we asked subjects from the U.S. the following question: How many sexual partners would you ideally like to have over the next thirty years?" The mode and average conditions received the following additional statement: "The [most frequent response/average] given by Heterosexual [Female/Male] subjects between [18 and 23/24 and 29] years was X1 sexual partner(s). A value of X2 sexual partner(s) was given by X3 of the subjects in this group." The value [most frequent response/average] depended upon the experimental condition; the values [Female/Male] and [18 and 23/24 and 29] matched the gender and age group of the participants. The values X1 and X2 were the averages and modes for desired number of sexual partners, and X3 was the percentage of the age/gender group that reported 1 partner.

The statistical information presented to participants in Study 3 was based on participants' reports in the first two studies. For instance, the mode condition for young females read as follows: "We previously ran a study on MTurk in which we asked subjects from the U.S. the following question: How many sexual partners would you ideally like to have over the next thirty years? The most frequent response given by Heterosexual FEMALE subjects between 18 and 23 years was one sexual partner. A value of one sexual partner was given by fifty-six percent of the subjects in this group." (See Table 2 for other statistical information presented in normative messages.)

At screening, subjects were asked about their age, gender, level of education, sexual orientation, and country of birth. At the end of the study, they were asked their current relationship status, race, their number of previous sexual partners, and political affiliation and orientation.

**Procedure.** Participants who consented to take part in the study and who met the screening criteria were first asked to estimate the distribution of the desired number of sexual partners of their male and female peers, separately, within their same age range. The format of the question was the same as in Study 2. Next, participants were randomly presented one of the three normative messages. The normative message described either the average or the mode (i.e., most frequent) desired number of sexual partners of their peers (same age range and gender), or a control message. Then, participants were asked the same demographic questions as in Studies 1 and 2. Last, participants were thanked and given a randomly generated survey code to redeem their reward for completing the survey, in its entirety.

## Results

**Effect of information norms.** Fig 3 shows the frequency of the number of desired sex partners reported by each combination of age, gender, and condition. The filled black squares report desired partners for participants in the control group; the purple circles report the average condition; and the green diamonds, the mode condition (11 participants across groups

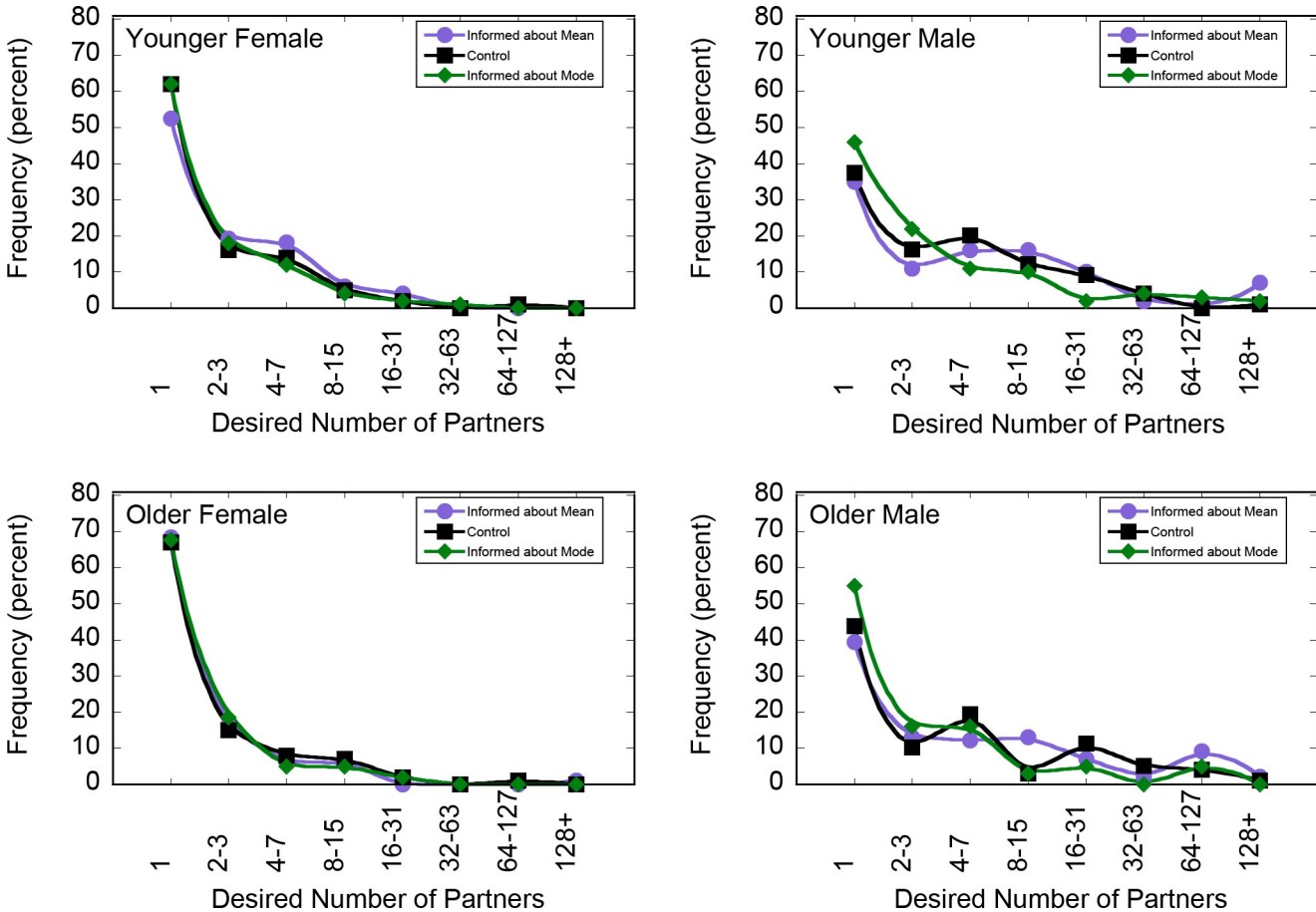

**Fig 3. Results of Study 3.** The frequency of reported number of sexual partners ideally wanted over the next 30 years by A) Younger Women, B) Younger Men, C) Older Women, and D) Older Men. Black squares report frequencies for subjects in the control condition, purple circles for the mean condition, and green diamonds for the mode condition.

desired no partners; their data are not depicted in the figure but are included in analyses). The panels represent the same gender and age groups as Figs 1 and 2. As expected from the previous studies, for all groups and all conditions, the modal number of desired sexual partners is 1, and the distribution has a long tail. The means, medians, and modes for desired number of partners for each age and gender group are presented in Table 1.

We are interested in whether the different information conditions affect the number of participants who report desiring a single sexual partner. Given the skew of the data, we examined the effect of the information condition on the desired number of sex partners using the Kruskal-Wallis test. Information condition affected desired number of partners ($\chi^2(2) = 10.75$, $p = .005$): participants desired fewer partners in the mode condition than in the no-information control ($\chi^2(1) = 5.344$, $p = .021$) or the average condition ($\chi^2(1) = 10.06$, $p = .002$). The effect of the information condition was statistically significant for men ($\chi^2(2) = 12.233$, $p = .002$) but not for women ($\chi^2(2) = 1.575$, $p = .455$).

The percentage of male subjects who desired fewer than four partners was higher in the mode condition (69.5%) than in the average condition (50.8%; $\chi^2(1) = 14.62$, $p < .001$) and the control condition (54.5%, $\chi^2(1) = 9.45$, $p = .002$). Similarly, the percentage of male subjects who desired more than 10 partners was lower in the mode condition (12%) than in the average

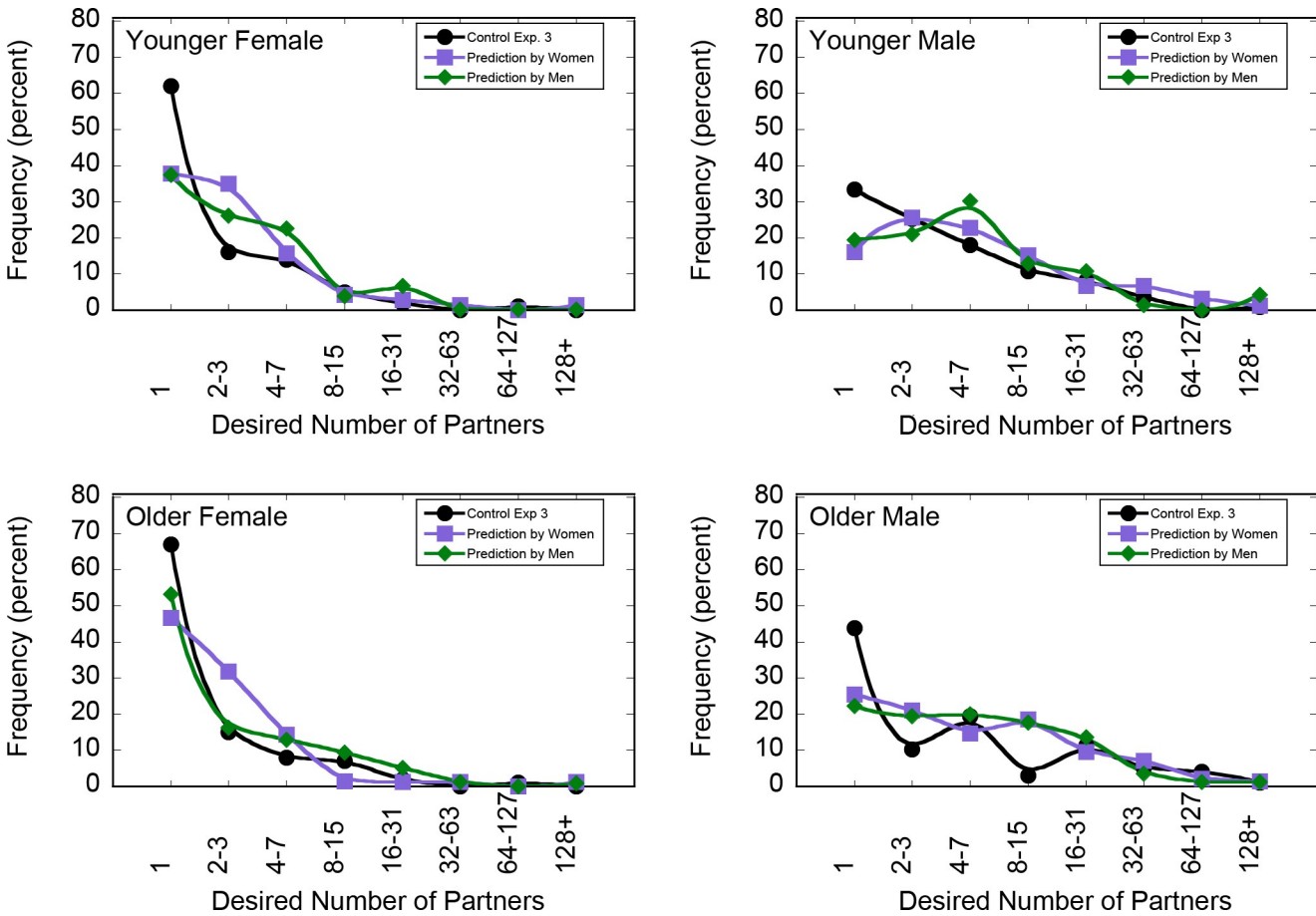

**Fig 4. Study replication of Study 2 using reports of the desired number of partners from the control group from Fig 3 and estimation of desired number of partners from individual reports from all conditions.** Symbols are the same as Fig 2. Two older males reported desiring no partners and are not presented in the graph, though they were considered in all analyses.

condition (24.1%; $\chi^2$ (1) = 9.91, $p$ = .002) and the control condition (20.2%, $\chi^2$ (1) = 4.96, $p$ = .026)

While the study's aim was to investigate the effect of information on desired number of sexual partners, we also asked participants to predict the reported number of ideal sexual partners; therefore, we can also report a replication of Study 2. As in the previous studies, those who thought fewer people wanted exactly one partner wanted more partners for themselves ($r_S$ (1196) = -.415, 95% CI [-0.463, -0.366], p < .001). Fig 4 plots distributions of responses to questions about own desired number of partners for the control group and predictions of others' desires. As can be seen from the figure, participants once again greatly underestimated how many people of their age, for both genders, would report desiring exactly one partner. Across age and gender groups, the average estimate was that 30.3% (95% CI [28.79%, 31.81%]) of same-gender, same-age people would desire exactly 1 partner, whereas from 37.37% to 67.00% of subjects in the control condition desired exactly one partner, depending on age and gender.

As with Study 2, to calculate the underestimation of the number desiring exactly 1 partner, we subtracted the estimated percentage for one's age and gender who desired exactly 1 partner from the percent who desired exactly 1 partner. The average *underestimate* was 22.2%, similar to the results of Study 2.

We again tested for differences in the misperception of the desired number of partners for men versus women. To test misperception, we used the same process as in Study 2. Once again, subjects underestimated the percentage of women who desired exactly one partner (mean underestimate 28.3%) more than they underestimated the percentage of men who desired exactly one partner (mean underestimate 18.0%), $t$ (1195) = 17.22, $p < .001$.

**Discussion.** Study 3 shows that male participants shifted the number of desired sexual partners depending on the normative information provided to them. The results indicate that social norms interventions may be useful for considering sexual attitudes. Furthermore, participants underestimated the number of others who desire one partner, replicating the results of Study 2.

## General discussion

In three studies, we asked participants to state their desired number of sexual partners over the next 30 years, and then we asked participants to make predictions about the responses of others. We have replicated the pattern of results from Pedersen et al. [44]; that is, the number of partners reported by men and women produced skewed distributions, with a mode of one desired sexual partner, and the skew is greater for men than for women. Our studies address three further questions: 1. How accurately do people estimate the modal number of sexual partners that others desire? 2. Does misunderstanding of this mode have social implications? 3. Is it useful to consider the mode, rather than just the mean, when working with skewed distributions?

### How accurately do people estimate the modal response of others?

In Study 1, we asked participants to estimate the modal number of sexual partners desired by men and women. As shown in Fig 1, participants' estimates of the mode were poor, and sometimes dramatically so. For instance, not a single young male participant estimated the modal response of younger men to be one sexual partner. The result creates the unusual situation where about 40 percent of young men reported desiring a single sexual partner, but not one of those men thought that they represented the most typical respondent. The mischaracterizations of the percentage of the population that desired one future partner also occurred for older men and both age groups of women.

One possible cause of this misestimation is that participants are simply misconstruing the meaning of "mode" in Study 1. We address this possibility in Study 2 by asking participants to report their estimate of the entire distribution of desired sexual partners. We therefore have two different ways of evaluating participants' perception of the mode: the direct estimate from Study 1, and the indirect estimate from the mode of the estimated distributions in Study 2. Study 2 showed that participants still underestimated the frequency with which others desired a single partner, though the estimate of the number of people desiring one partner was the highest value of the averaged responses. The misperception of the mode therefore is likely due partially, but not solely, to misinterpretation specific to the concept of the mode, and that misperception may be most common when the question about the mode is explicit. The differences between the methods highlight the advantages of having people estimate the whole distribution rather than just a particular statistic (see also [17, 51]).

There are other candidate mechanisms for why people misrepresent non-symmetric distributions such as that for desired number of sex partners. For instance, Nisbett and Kunda [15] reported a bias toward estimating symmetric (or normal) distributions. Dannals and Miller [19] account for this bias by noting the difficulty people have in understanding the effects of outliers and attributed this to a form of representation error [5, 52]. In Dannals and Miller's proposal, when people are estimating a distribution, exemplars of men or women who desire a

larger number of partners are perceptually more salient and therefore stand out and lead to a skew in estimation that is much larger than in the data.

## Does misunderstanding of the mode have social implications?

Social norms interventions–that is, giving people accurate information about population distributions of peer behavior–can correct misperceptions and change behavior [30, 31]. In Study 3, prior to questionnaire completion, we informed participants of the mean or mode from the respondents in Studies 1 and 2. Presenting accurate information about the mode lowered the number of sexual partners that Study 3 male respondents said they desired but there was no evidence of an effect on women's reports. In contrast, there was no evidence that providing participants with accurate information about the means affected the number of desired sexual partners for Study 3 respondents. We are aware of no other studies comparing the effects of sharing these two forms of normative information. Future social norm interventions might consider the use of information about the modal versus mean experiences of others.

The misestimation of the predicted desired number of sexual partners may have a number of social consequences since, on average, participants who think others desire more partners report wanting more partners themselves. Since the provision of accurate information about modes led men in Study 3 to report desiring fewer partners, it is possible that the misperception we identify here leads some to report desiring more partners than they would choose absent the misperception, and to judge themselves peculiar for their preferences for a low number of partners.

Some limitations should be kept in mind when considering the implications of this work on the misestimation of desired number of sexual partners. We drew our participants from the crowdsourcing tool Amazon Mechanical Turk and limited the sample to self-reported heterosexuals to maintain sufficient sample size and homogeneity. It seems possible that the sorts of processes that would lead to the misperceptions would generalize to other sampling procedures (and other groups), but this needs to be empirically verified. While we had no way of identifying individual participants and assured them of their anonymity, it is still possible that their reports of their own desires were influenced by social desirability, so participants may have reported more or fewer desired partners than they wanted so as to maintain appearances [53]. Our item seeking participants' desired number of partners used the phrase "ideal" so as to be consistent with previous important research on this topic (e.g., [43, 44]). It is possible that different ways of asking about the desired number of partners would yield different answers. Future longitudinal research should examine the ability of current illusions regarding social norms to predict subsequent behavioral, social, and affective consequences. This is particularly important for the behavior following presentation of social norms messages, such as those used in Study 3, to preclude transitory demand effects. Does shifting the reported number of partners, as happened in Study 3, shift subsequent behavior?

Our normative data are descriptive, not prescriptive. We report the proportion of respondents in our sample who said they desired a given number of partners, not how many partners people *should* desire. However, our subjects clearly misperceive the desires of others, and it seems likely this misperception shaped their own reported desires.

## The conflict between the mean and mode

While our data speak to the misperception of others' preferred numbers of sexual partners, the data also speak to the more general problem of how people (mis)understand skewed distributions, and to the importance of considering the mode when distributions are skewed. The current studies can be considered analogous to a framework that one of the authors has applied to

his research on visual illusions, in which illusions arise not only from "differences between perception and reality" but also from "conflicts between possible constructions of reality" [2, 3]. In the current studies, conflict arises because the mean and the mode convey different information, and like many visual illusions, the perception of events will depend on which type of information is given more weight.

In our sample, it is true that the average number of sexual partners reported by men is higher than the mean number reported by women, but it is also true that in many of our conditions, half of the men reported numbers that were less than the average number reported by women (that is, the men's median was less than the women's average). When reporting our data, then, one could stress either the average of the distributions, in which case men desire a significantly higher number of partners than do women, or one could talk about the mode of the distribution, in which case men and women report similar desires. The same dataset, therefore, could be used to support the opposing claims that men and women have similar (mode) or different (mean) sexual desires.

Most behavioral studies choose simply to report the differences between the means and are not concerned with the mode or with other measures of centrality [54]. This is not without reason. The mean is a methodologically convenient statistic: the central limit theorem allows us to assume that for a big enough sample size, the average of our sample will be normally distributed, and this allows for a wide variety of inferential comparisons. However, the literature contains numerous accounts of the problematic nature of hypothesis testing with the mean, including the problems that arise when distributions are not symmetric [24, 25, 55]. The mode, on the other hand, to use Chacon's [21] phrase, seems like a "bizarre" statistic that is highly variable and allows only a limited number of inferential methods. Therefore, even though different sorts of information may be provided by the mode (or other measures of centrality [54], such statistics are often simply not reported.

Here we illustrate a type of problem that–though obvious–may be worth emphasizing since it is directly relevant to the interpretation of the data and seems to be under-discussed in the literature. That is, a difference between means can indicate that distributions *are* different from each other; they cannot indicate *how* the distributions are different from each other. For example, in Fig 5, the panels on the left show two different models underlying differences between means. The panel on the upper left (A) illustrates an independent variable that affects all the members of the control group: if the blue line is the distribution of the control group, then the effect of the independent variable is to shift the whole distribution to the right (red line). That is, if N ($\mu$, $\sigma$) is a normal distribution with mean $\mu$ and standard deviation $\sigma$, then the blue line = N (4,1.5), and the red line = N (5.5,1.5). The panel on the lower left (C) shows the condition where the independent variable affects only a portion of the population. For this figure, the blue line is the same as the control in panel A, and the red line shows a case where only 25% of the group are affected by the treatment; that is, red line = 0.75*N (4,1.5) + 0.25*N (10,1.5), and then is normalized so that the area equals 1.0.

In terms of the population means, the effects of the treatment vs. control are identical in Fig 5A and 5C. That is, in both figures, the blue curves have a mean of 4.0 ($\mu_{control1}$ = 4.0), and the red curves have a mean of 5.5 ($\mu_{treatment}$ = 5.5). The samples from these distributions would not be distinguishable from each other except for a slightly larger confidence interval for samples from the red curve in Fig 5C. The right-hand plots (panels B and D) show the 95% confidence interval of the means from 300 random samples (MATLAB-generated) from each distribution. The confidence intervals are centered around the means, and while the interval is larger for the skewed distribution in panel C, inference testing would clearly find the means to be significantly different from each other (MATLAB gives independent-groups t-test probabilities for the samples in panels B and D of p = 1.75e-63 and p = 1.34e-27).

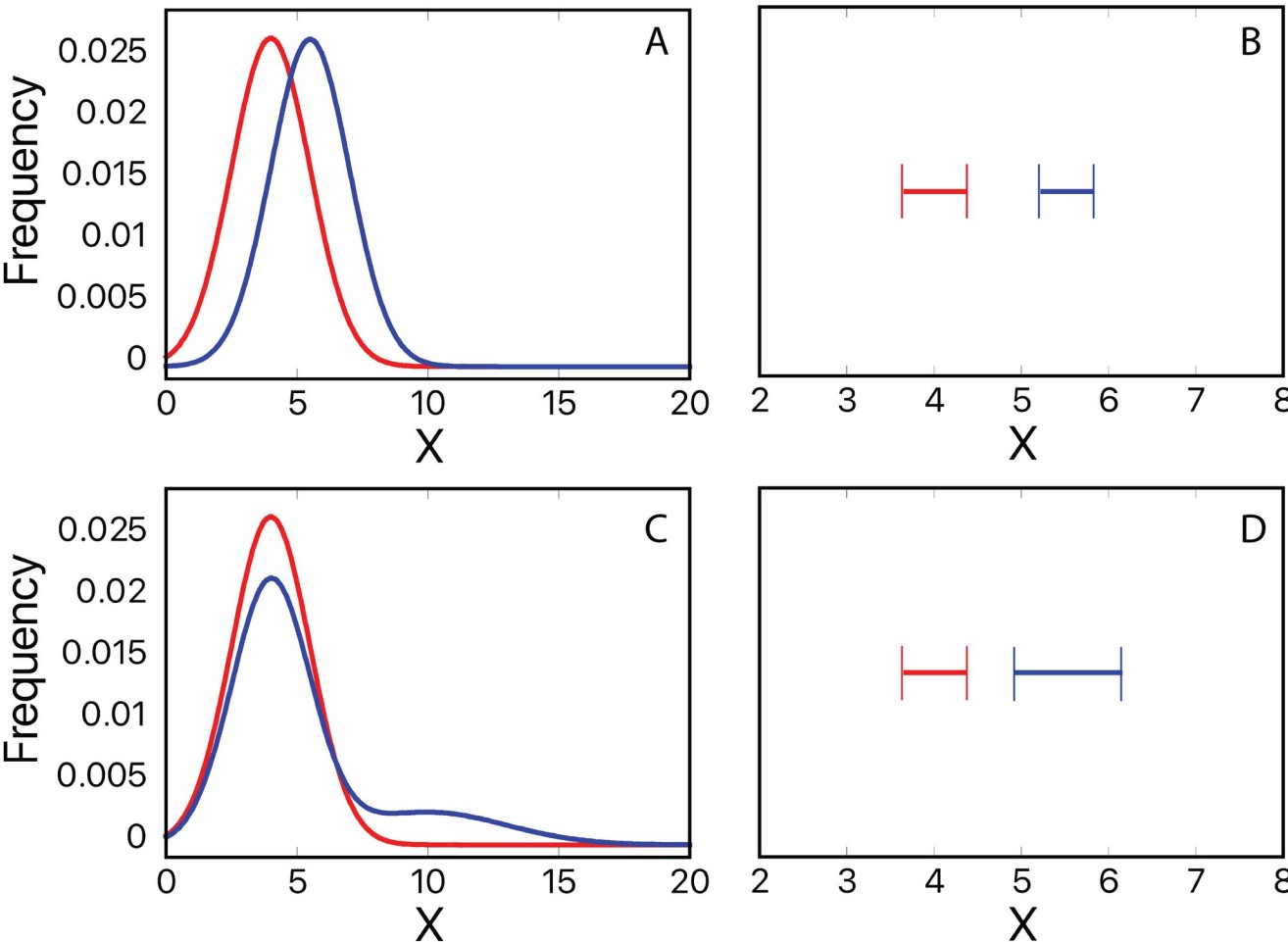

**Fig 5. Differences between means can indicate that distributions *are* different from each other; they cannot indicate *how* the distributions are different from each other.** A) Illustrates the textbook model in which the independent variable affects all the members of the control group: control-group distribution (blue line), treatment-group distribution (red line). B) 95 percent confidence intervals of 300 random samples from distributions in A: control-group (blue line); treatment-group (red line). C) Illustrates condition where the independent variable affects only a portion of the control group population. D) 95 percent confidence intervals of 300 random samples from distributions in C. If an experiment only reports a significant difference between the means (like the confidence intervals on the right), then that experiment cannot differentiate an effect like that shown in panel A from an effect like that shown in panel C.

Here, then, is the problem: just as summary statistics can arise from wildly diverse types of data samples (see Anscombe's quartet [56] and Alberto Cairo's "Datasaurus" [57]), significant differences between sample means can arise from diverse forms of population distributions. So, if an experiment only reports a significant difference between the means (like the confidence intervals on the right of Fig 5), then that experiment cannot differentiate an effect like that shown in panel A from an effect like that shown in panel C. But these two effects are substantially different from each other; the effect in panel A is what is typically presented when illustrating that "group A is different from group B," whereas the effect in panel C is what would happen if the independent variable affected only a portion of the population. The panel C effect is similar to a story related to one of us by a colleague in the pharmaceutical industry, who describes a drug under review as "the drug [that] didn't work for most people, but for those it did work for, it had a whopping effect."

In our study, the model in panel A is like the statement, "Men desire more sexual partners than women," and the model in panel B is like the statement, "Overall, men and women report similar desires, but a relatively small percentage of men report desiring a large number of partners." Indeed, in our samples, the mode is always one, and the differences between the male and female distributions arise primarily because of a small cohort of men that report desiring a higher number of partners. Across our studies (using only the control group from Study 3, not the experimental groups), 3.3% of men (17 of 513) wanted more partners than the highest number reported by any of the women (60), with six of the men desiring at least 240 partners.

We have no insight as to the characteristics of the cohort that creates the long tail of the male distribution, but it is easy to generate hypotheses about their origin. For instance, perhaps the cohort includes a few men who (a) are successful at obtaining multiple partners and are just projecting forward; or (b) have had few sexual partners but think that a large number of partners would satisfy their social status goals; or (c) are simply responding to the demand characteristics of the study and report a value that accords with how they think they should respond. While distinguishing between such potential causes is beyond the scope of this study, it does seem that there would be both interest and social value in understanding whether there are indeed separable cohorts of the male population in this regard.

While an analysis of means alone cannot distinguish a shift (Fig 5A) from a skew (Fig 5C), there are certainly several methods that can distinguish between the distributions. For instance, in Fig 5A, the two curves have different values for mean and mode, whereas in 4B, the red and blue curves have different means, but the same mode; therefore, an examination of the mean and mode would be able to distinguish between the conditions. There are numerous procedures for analyzing the amount of overlap between samples. For instance, there are methods of examining the differences between distributions, such as the Kolmogorov–Smirnov test, other methods based on Kullback–Leibler divergence, and a variety of non-parametric distribution-free effect-size measures that could differentiate between distribution overlap (techniques were recently reviewed [58]).

Lastly, we wonder whether similar mischaracterizations may arise in other behavioral studies, and whether this has any effect on the reproducibility crisis that is currently a major topic in the field [59, 60]. If a particular behavioral study had an independent variable that affects only a small cohort of the treatment group, then the size of that cohort would drive the presence or absence of statistical significance in the experiment. This would mean that 1) there are conditions where an experimenter concludes that group A is different from group B, but really only a cohort of group A differs from group B, or the experimenter concludes that "people are subject to cognitive illusion X," when a statement such as "a segment of people are subject to cognitive illusion X" is closer to reality; 2) the extent to which an experiment is reproducible depends only upon a segment of the cohort, which could be caused by inherent inhomogeneity within the cohort or by other experimental/individual interaction such as demand characteristics [61]; and 3) it may be harder to identify the effect of small cohorts in multivariate studies since such studies often have procedures that only identify differences between means, and the effect of skewed distributions would be harder to identify in large tables of numbers (although this problem may be alleviated by improved procedures for visualizing multivariate data). Thus, it seems to us that including analyses of other measures of centrality may improve many aspects of psychological science. This view is consistent with that of others (see [26], for a review of critiques of "averagarian" approaches to behavioral measurement).

## General conclusion

We asked men and women how many sexual partners they desire over the next thirty years and then asked them to make estimates of these data. When asked directly, study participants were remarkably poor at estimating the most frequent number of sexual partners; when asked to estimate the shape of the whole distribution, participants performed more accurately but were still prone to underestimate the proportion of participants who reported one sexual partner (the mode). We do not believe the misestimations made by participants have to do with the nature of the question per se (i.e., it is not that the question has to do with sex); rather, the pattern of errors should be expected given that participants were estimating the parameters of skewed distributions. We would therefore expect similar patterns with other issues of high social relevance that also have skewed distributions (for example, daily performance of stocks that compose financial market indexes, household income, or the presence of anxiety).

The skewed shape creates two true–but seemingly contradictory–statements about the desires of men and women: 1. On average, men would ideally like to have more sexual partners than women (men's average from studies 1 and 2 plus the control condition of Study 3 is 73.8, whereas women's average is 3.23); and 2. The most frequent response given by *both* men and women is that they would like to have only one future sexual partner. The apparent conflict between the two statements arises because skewed distributions create differences between the mean and the mode: the means of the men's responses are higher than the means of the women's responses, but the modal responses of men and women are the same. Hence, when men are described as wanting more sexual partners than women [43], more weight is being given to the small proportion of men who report large numbers than to the roughly 40% who report one partner. So, while a statement such as "On average, men want more sexual partners than women" is technically true, it fails to represent the responses of roughly 40% of the men in our studies, who state that ideally they would like just one partner.

Lastly, we stress that the differences between the mean and the mode in our data are part of larger issues concerning how standard statistical techniques can create misleading interpretations of data. It is well known that summary statistics can hide trends. For instance, Alberto Cairo's Datasaurus Dozen [57] shows that the same mean, standard deviation, and covariance can result from data that, when plotted, look like a star and from data that look like a dinosaur. Similarly, techniques intended to discern differences between means are also ambiguous to the underlying distributions (in terms of the Datasaurus dozen, these techniques may be able to identify a difference between the means of the star distribution and dinosaur distribution, but they still will ignore the shape of the distribution). To this end, we created a simple demonstration (Fig 5) to show that techniques intended to find difference between means cannot differentiate between shifts in a small cohort of the population (like those produced in our data) and those produced by shifts in the whole distribution. It therefore seems likely to us that the current reproducibility crisis is exacerbated by the lack of statistical techniques that differentiate between cohort shifts and population shifts. This potential issue may be alleviated by the judicious use of analyses that include estimates of centrality besides the mean.

## Acknowledgments

**Author notes**

Arthur G. Shapiro and Anthony H. Ahrens, Department of Psychology, American University. Rubie M. Peters, Department of Psychology, University of Mississippi. Study 1 and methodology of Studies 2 and 3 come from the master's thesis research of Rubie M. Peters, fulfilled at American University and completed while at the University of Mississippi. The authors thank Scott Parker and Betty Malloy for valuable discussions about the paper, particularly

regarding material related to Fig 5, and we thank Sherri Geller for assistance in editing the manuscript.

## Author Contributions

**Conceptualization:** Arthur G. Shapiro, Anthony H. Ahrens.

**Data curation:** Arthur G. Shapiro, Rubie M. Peters, Anthony H. Ahrens.

**Formal analysis:** Arthur G. Shapiro, Rubie M. Peters, Anthony H. Ahrens.

**Investigation:** Arthur G. Shapiro, Rubie M. Peters, Anthony H. Ahrens.

**Methodology:** Arthur G. Shapiro, Rubie M. Peters, Anthony H. Ahrens.

**Project administration:** Arthur G. Shapiro, Rubie M. Peters, Anthony H. Ahrens.

**Resources:** Arthur G. Shapiro, Rubie M. Peters, Anthony H. Ahrens.

**Software:** Arthur G. Shapiro, Rubie M. Peters.

**Supervision:** Arthur G. Shapiro, Anthony H. Ahrens.

**Validation:** Rubie M. Peters.

**Visualization:** Arthur G. Shapiro, Rubie M. Peters, Anthony H. Ahrens.

**Writing – original draft:** Arthur G. Shapiro, Rubie M. Peters, Anthony H. Ahrens.

**Writing – review & editing:** Arthur G. Shapiro, Rubie M. Peters, Anthony H. Ahrens.

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
