## [Decision Letter · Decision Letter 0]

5 Jul 2024

PONE-D-23-30826(Mis)estimation of the modal number of desired sexual partnersPLOS ONE

Dear Dr. Shapiro,

Thank you for submitting your manuscript to PLOS ONE. After careful consideration, we feel that it has merit but does not fully meet PLOS ONE’s publication criteria as it currently stands. Therefore, we invite you to submit a revised version of the manuscript that addresses the points raised during the review process. Thank you for submitting your manuscript to PLOS One. It is an interesting topic. The article is well written and has a nice logical flow. The reviewers have issues as noted in their reviews so please respond to their concerns. If you could include just a bit more about how to address the confusion among the measures of central tendency and how to teach our students to recognize that and initially check for normality so that we select the correct stats.

We look forward to receiving your revised manuscript.

Kind regards,

Mary Diane Clark, PhD

Academic Editor

PLOS ONE

Journal Requirements:

**Additional Editor Comments:**

Thanks for your patience with the length of time of this review. It has become difficult to get reviewers at this time but I appreciate your understanding.

You have two different reviews and I think that both have important contributions. Reviewer 1 challenges you about the common issues with people not checking for normality. Reviewer 2 has fewer issues but some important issues. It was difficult to pick between minor and major review, but please respond to the comments from each reviewer.

I have several easy comments. 1--check all of your spacing as you have some issues with that. 2--you do not need statistics in your discussion. Please remove those.

Reviewers' comments:

Reviewer's Responses to Questions

**Comments to the Author**

1. Is the manuscript technically sound, and do the data support the conclusions?

Reviewer #1: Partly

Reviewer #2: Partly

2. Has the statistical analysis been performed appropriately and rigorously? 

Reviewer #1: N/A

Reviewer #2: Yes

3. Have the authors made all data underlying the findings in their manuscript fully available?

Reviewer #1: Yes

Reviewer #2: Yes

4. Is the manuscript presented in an intelligible fashion and written in standard English?

Reviewer #1: Yes

Reviewer #2: Yes

5. Review Comments to the Author

Reviewer #1: The topic of the manuscript is interesting, but revision is needed in all sections:

Introduction

The authors seem to wish to talk about two separate topics: perceptual illusions and use of the arithmetic mean to provide the average of a data set. Although the two topics are related in their study, the Introduction fails in telling the reader why this is so. In fact, the issue of the arithmetic mean is not handled well by the authors (see Discussion).

Study 1

The X-axis in the figures is very confusing. Were you trying to use a logarithmic scale so that a power function becomes a straight line? If so, this is not clear. Also, the resulting curves do not approximate straight lines. Wouldn’t a linear axis be just as good while being much easier to understand?

Is the mean for men really greater than 500? This does not look right.

Study 2

This is very confusing. Strong rewriting is recommended.

Study 3

Please clarify why the reader should be impressed by the fact that, after being told what the expected answer would be, the participants tended to give the expected answer.

Discussion

The authors make a big deal out of the fact that the mode can be different from the arithmetic mean. Any reasonable researcher looks at the data before running any type of analysis. In the case of these data, it is evident that the distribution is skewed. This will cause the researcher not only to choose the mode (or median) over the mean but also to remember that most statistical tests (and definitely the ANOVA) assume that the data are normally distributed (in which case the mean and the mode are the same) and that the result of the test is not reliable if the data don’t pass the normality test.

An interesting topic for discussion is whether people who are not trained in statistics have a bias towards the mean. A different set of questions would be needed, however, to differentiate a bias towards the mean from a bias towards normal distributions. And, of course, one could argue that bias is not the correct word because a preference for normal distributions derives from the fact that most distributions are normal (height, weight, intelligence, beauty, etc.).

As for the main research finding that most people would ideally like to have one partner for life even though they (especially men) have many more, did the authors consider that the adverb "ideally" introduced a fundamental bias in the answers given by the participants? In a country where polygamy is illegal (even if serial monogamy is silently accepted), an ideal relationship is expected to be singular.

Reviewer #2: This manuscript presents a successful replication of a 2002 paper reporting a mis-estimation effect for participants estimating the modal number of preferred sexual partners. A second pseudo-replication study employing altered participant instructions investigates a possible explanation for the effect (misunderstanding of the question posed), and a third study tests the effect of exposure to information about the central tendency in a previous sample on participant reports. I very much enjoyed reading this manuscript, and in particular thought the relation of the reported results to the general underlying issue of reported means potentially concealing different distributions (e.g., in fig 5) very well-handled and potentially useful. The standard of writing is excellent and the arguments are clearly presented throughout. Besides noting the lack of justification of sample size, I have only some comments regarding the presentation of results.

1) Some justification of sample size will need to be provided to meet journal requirements (https://journals.plos.org/plosone/s/criteria-for-publication). In Study 1, for example, it is stated that an aim was to collect a “sufficiently large” sample, but there is no details are provided (e.g., what effect size the sample would be sufficient to detect).

2) In some of the results and discussion, non-significant p values appear to be interpreted as evidence for null results. For example, p.11, line 11 “There were no differences in the mean rank of predictions …p = .053”. A non-significant p value cannot be interpreted (in isolation) as evidence for the null because no distinction is made between evidence for not effect and there simply being no evidence for an effect (i.e., because the sample is not large enough to detect an existing effect). I recommend rephrasing (in the results and discussion) any interpretation of non-significant p values which could be taken to suggest a claim of evidence for the null. For example, p.12 “Men and women’s predictions do not differ from each other”. While this might be read as "there was no evidence that men and women's predictions differ", as phrased it could be interpreted as a claim that men and women's predictions do not differ (and they may well differ in a larger sample, if the study is under-powered to detect an existing difference). Alternatively, claims of evidence for no difference could be tested using an appropriate method such as equivalence testing or Bayes factors, e.g., see https://doi.org/10.1093/geronb/gby065.

3) P.11, line 5 “only a marginal effect of gender”

The p value is .079. At a conventional alpha of .05, both this and the smaller p value of .053 reported further down the same page these are non-significant.

4) I think it would be useful to report confidence intervals alongside point estimates (e.g., for correlation coefficients and mean differences).

5) Data are only available in a proprietary format (SPSS .sav files). For accessibility I suggest also uploading data files as .csv files.

6. PLOS authors have the option to publish the peer review history of their article (what does this mean?). If published, this will include your full peer review and any attached files.

Reviewer #1: No

Reviewer #2: No

---

## [Author Response · Author response to Decision Letter 0]

15 Oct 2024

September 22, 2024

Dear Professor Clark,

We thank you for your time and effort editing our manuscript. We have responded below to comments from the two reviewers. We thank both reviewers for their efforts and believe they have contributed to an improved paper. 

As you mentioned in your summary statement: “Reviewer 1 challenges you about the common issues with people not checking for normality and Reviewer 2 has fewer issues but some important issues.” 

We think that our phrasing may have led Reviewer 1 to a simplistic view of our thesis because of some of our phrasing. We have tried to correct this misapprehension by modifying the stress of one paragraph in the introduction and one paragraph in the discussion. The two major issues were, as the editor summarized, “don’t people just check for normality,” and why are we “making a big deal out of the fact that the mode can be different from the arithmetic mean.” As to the first point, it is hard to check normality on multivariate studies with many cells (although this process is improved with violin plots), but even if a researcher finds nonnormality, what do they do about it? Often researchers are not concerned about normality because the central limit theorem shows that the sample means will be an unbiased estimate of the statistic in which they are interested ( we cite specific text from an SPSS manual in our response to specific comment below but one of us taught intro stats for twenty years, and he could find many more examples). Our figure 5, therefore, highlights a different approach to the problem that arises when methodologically, researchers are not concerned with the shape of population distributions. 

But more importantly, the section related to Figure 5 is not simply about normality or about the difference between the mean and the mode. We are trying to stress that just as summary statistics can arise from wildly diverse types of data samples (see Anscombe's quartet (Anscombe, 1973) and Cairo’s “Datasaurus” (Matejka & Fitzmaurice, 2017)), significant differences between sample means can also arise from diverse forms of population distributions. Anscombe and Cairo’s Datasaurus attract (justly) a lot of attention and are often used as lead-ins to data science talks. Oddly, our figure five may be the first paper to show that these ambiguities also apply to comparisons between summary statistics. We think that, while not entirely shocking when pointed out, is an important addition to the field and is directly germane to interpretations of our experiments. 

Reviewer two had important suggestions about the analysis of the data. We have responded to these suggestions and have done a thorough reanalysis of the data in the paper. In some instances we found very small and inconsequential discrepancies from our previous draft. We have substituted the new numbers. 

Lastly, we have checked the spacing on the document, and have removed the statistic in the discussion, as per your suggestion. 

 Sincerely,

 Arthur Shapiro, Anthony Ahrens, and Rubie Peters

Response to Reviewer 1

Introduction

The authors seem to wish to talk about two separate topics: perceptual illusions and use of the arithmetic mean to provide the average of a data set. Although the two topics are related in their study, the Introduction fails in telling the reader why this is so. In fact, the issue of the arithmetic mean is not handled well by the authors (see Discussion).

We thank the reviewer for this comment. We think it is a misreading to say that we are or may be seeking to draw a comparison between the illusions and the “use of the arithmetic mean to provide the average of a data set.” We see how the reader could have come to this interpretation, particularly if the reader were taking their cue from the first sentence of the final section–which overemphasized the “mean.” We therefore revised the leading sentence to read “The problem of using a single summary statistic to describe the central tendency of skewed distributions is illustrated in the distribution for survey responses to the following question” By removing the word “mean” from that topic sentence we hope that others will not be misled. 

The other part of the issue has to do with the definition of “illusion.” Colloquially, people talk about illusions in terms of a difference between perception and reality. More recently, the Shapiro lab has been part of a discussion concerning the utility of other frameworks for considering visual illusions. One such framework is that illusions arise not only from “differences between perception and reality” but also from “conflicts between possible constructions of reality” (Shapiro & Hedjar, 2019; Shapiro, 2021). As we note in the introduction and in the discussion, in the studies presented here, the conflict arises because the different measures of central tendency convey different information, and like many visual illusions, the perception of events will depend on which type of information is given more weight. 

Study 1

The X-axis in the figures is very confusing. Were you trying to use a logarithmic scale so that a power function becomes a straight line? If so, this is not clear. Also, the resulting curves do not approximate straight lines. Wouldn’t a linear axis be just as good while being much easier to understand?

Thank you for the comment. The scale reflects the idea that human scaling of numbers is closer to a Fechner (that is, logarithmic) response function than to linear response function. An observer is more likely to differentiate between 1 vs 2 than they are to differentiate between large numbers, say, 523 vs 524. The curves do not approximate straight lines, nor were they designed to do so. Rather, the curves show that the distributions of judgments are skewed, with particular and systematic differences between observed and expected frequency. In addition, in studies 2 and 3, participants were asked to give percentages of people who desired a given number of partners (e.g., 32-63). Thus, for figures 3 and 4, the x-axis displays the exact range of possible desired partners for which participants were to give estimates. We believe there is some advantage to using similar scales for estimates for the three studies. 

Is the mean for men really greater than 500? This does not look right. 

We have double checked, and the mean is correct. There were two outliers in this data set. One younger male reported desiring 20000 partners, and one older male reported desiring 10000 partners. No one else wanted more than 300 partners. These two participants raised the mean. This is part of the reason that we report a Kruskal-Wallis test as well as an ANOVA. We have added this point about extreme responses on p. 12. 

 Study 2

This is very confusing. Strong rewriting is recommended.

Thank you for this comment. We have made some changes to the introduction of study 2, removing the term “innumeracy,” which we might have caused confusion. We have balanced this reviewer's response to that of the second reviewer, who wrote, “The standard of writing is excellent and the arguments are clearly presented throughout,“ and with the comments of other reviewers who gave us feedback on this section. 

Study 3

Please clarify why the reader should be impressed by the fact that, after being told what the expected answer would be, the participants tended to give the expected answer.

We are uncertain of the reviewer’s concern here. We did not tell participants what the “expected” answer was. Rather, consistent with other research using social norms interventions, we gave participants information about others’ desired number of partners. Perhaps the reviewer is concerned that the experimental results (with conditions affecting reported number of partners) are due to demand characteristics. This is an important point that holds for much of social norms intervention literature. We had, indirectly, addressed this point by recommending that future research include behavioral data. We have made this explicit by adding “so as to preclude transitory demand effects” to our discussion of the need for behavioral data on p. 27. 

Discussion

The authors make a big deal out of the fact that the mode can be different from the arithmetic mean. 

Thank you for this comment. The response indicates to us the importance of the inclusion of Figure 5 (and accompanying text) in the document. The standard way of presenting data in the psychological literature is as averages (i.e, means) with confidence intervals. The main argument of this section is not that mean and modes are different in skewed distributions, but rather that the standard methods of summarizing data are inherently ambiguous as to the shape of the underlying population distribution. While this point, too, should be obvious, we think that our method of presenting (i.e, figure 5) is unique, and points to an important issue in the psychology literature and is directly relevant to the interpretations of our experiments. 

“Any reasonable researcher looks at the data before running any type of analysis. This will cause the researcher not only to choose the mode (or median) over the mean but also to remember that most statistical tests (and definitely the ANOVA) assume that the data are normally distributed (in which case the mean and the mode are the same) and that the result of the test is not reliable if the data don’t pass the normality test.”

We have two responses: 1. Do researchers always look at their data? Certainly, most do for small studies, but we would be surprised if this is the case for multivariate studies, particularly with very large data sets (the movement towards violin plots in the neuroscience literature is a noted improvement). 

2. Even if researchers do find skewed sample distributions, what are they supposed to do? Most researchers are interested in finding a comparison between the summary statistics (are means of the sample distributions different from each other?); methodologically, they are not interested in what is going on with the population distributions. 

See, for instance, the SPSS guide for the Shapiro-Wilk Test (https://www.spss-tutorials.com/spss-shapiro-wilk-test-for-normality/). 

This guide has a section titled “Limited Usefulness of Normality Tests” in which the author states, “So if sample sizes are reasonable, normality tests are often pointless. Sadly, few statistics instructors seem to be aware of this and still bother students with such tests.” The reason for this comment is that because of the central limit theorem, “many test results are unaffected by even severe violations of normality.” 

The SPSS guide is not alone. Most commentary on normality tests has some claim about not needing to know the shape of the population distribution in order to estimate a population parameter. Our section related to figure 5 is a way to show that confidence intervals are ambiguous as to shape underlying population distribution. So, as we mention in the section, the same way that Datasaurus shows that summary stats may hide important trends, the differences between summary stats shown in the standard techniques can also hide important trends. 

Lastly, this idea is important to us because it has led to a new line of research in one of our laboratories. We find it remarkable that such ambiguities are not being discussed in relation to the replication crisis in psychology. 

An interesting topic for discussion is whether people who are not trained in statistics have a bias towards the mean. A different set of questions would be needed, however, to differentiate a bias towards the mean from a bias towards normal distributions. And, of course, one could argue that bias is not the correct word because a preference for normal distributions derives from the fact that most distributions are normal (height, weight, intelligence, beauty, etc.).

The question of the degree to which people who are not trained in statistics in general have a bias toward the mean is potentially of interest. But as the reviewer notes, we would need to do a different set of studies to test that proposition. In our paper, we raise the topic of bias toward normal distributions as one of several explanations for relatively weak judgments about distributions that are not normal. In particular, we raise this topic when we describe a 1985 paper by Nisbett and Kunda, and work by others trying to understand what they found. We are fairly agnostic about whether the Nisbett/Kunda formulation of bias toward normality is the process underlying the biases our paper found. We considered adding further discussion of this idea, but in our judgment it would take us beyond the scope of the paper. We note also that we are unsure if most distributions are, in fact, normal. It is hard to know the universe of distributions, and so it is hard to know the distribution of distributions. In the paper we note that some important distributions are not normal. That there are such distributions seems to us a good reason to explore how people think about those non-normal distributions regardless of whether they are the most common form of distribution.

As for the main research finding that most people would ideally like to have one partner for life even though they (especially men) have many more, did the authors consider that the adverb "ideally" introduced a fundamental bias in the answers given by the participants? In a country where polygamy is illegal (even if serial monogamy is silently accepted), an ideal relationship is expected to be singular.

We thank the reviewer for this observation. We used the word “ideally” so as to be consistent with previous research on this topic (e.g., Buss & Schmitt, 1993). It is possible that other ways of wording the topic would yield different judgments of how many sexual partners a given person would desire. We have added this point in our discussion on p. 27.

Response to Reviewer 2

This manuscript presents a successful replication of a 2002 paper reporting a mis-estimation effect for participants estimating the modal number of preferred sexual partners. A second pseudo-replication study employing altered participant instructions investigates a possible explanation for the effect (misunderstanding of the question posed), and a third study tests the effect of exposure to information about the central tendency in a previous sample on participant reports. I very much enjoyed reading this manuscript, and in particular thought the relation of the reported results to the general underlying issue of reported means potentially concealing different distributions (e.g., in fig 5) very well-handled and potentially useful. The standard of writing is excellent and the arguments are clearly presented throughout. Besides noting the lack of justification of sample size, I have only some comments regarding the presentation of results.

1) Some justification of sample size will need to be provided to meet journal requirements (https://journals.plos.org/plosone/s/criteria-for-publication). In Study 1, for example, it is stated that an aim was to collect a “sufficiently large” sample, but there is no details are provided (e.g., what effect size the sample would be sufficient to detect). 

We thank the reviewer for this point. We need enough participants to get a reasonably good estimate of the mode. That the modal number of desired partners is 1 across all 12 of the groups (3 (study) x 2 (age) x 2 (gender)) gives some confidence that we have correctly identified the mode. We also needed to know the degree to which participants underestimated how many people would choose the mode. Therefore, the key statistic in the first study is that only 9.8% of participants correctly identified 1 as the mode. 

In the two places in which we mention taking steps to have a “sufficiently large” sample, the context is our decision to limit our sample to heterosexual individuals. For instance, in the first study we had 267 heterosexual participants but only 17 who identified as gay/lesbian. There was only 1 older gay male, and having a single participant in this category precluded being able to describe a

---

## [Decision Letter · Decision Letter 1]

10 Nov 2024

PONE-D-23-30826R1(Mis)estimation of the modal number of desired sexual partnersPLOS ONE

Dear Dr. Shapiro,

Thank you for submitting your manuscript to PLOS ONE. After careful consideration, we feel that it has merit but does not fully meet PLOS ONE’s publication criteria as it currently stands. Therefore, we invite you to submit a revised version of the manuscript that addresses the points raised during the review process. One of the reviewers gave you an accept and I also reviewed the paper.  I have a few minor editing issues I need you to fix. If you can take care of these I can recommend publication.

We look forward to receiving your revised manuscript.

Kind regards,

Mary Diane Clark, PhD

Academic Editor

PLOS ONE

Journal Requirements:

Additional Editor Comments:

before I sent the paper out for review, I reviewed it and found just a few things that I need yo to fix. Plus One doesn't have an editor so I need you to clean up just a few things for us

it should only take about an hour or two--here are my issue that need cleaned up:

Page 5 end of first partial paragraph THE median and mean

Page 5 u (e.g., educational tests; Ho& Yu, 2015)

In your marked up version on page 5 the sentence makes no sense to me there is something wrong there

Both in terms OF. The OF seem like it is needed

Then I can’t find what the both is

Seems like y our Galton quote needs a page number

End of page 6 The present study is an orphan header and need two lines of text below it

Same with Method at the end of the next page

In study one why did you use an ANOVA when you knew the distribution was skewed. Also, with numbers this large Field states that most of them have already approached normal ---

Page 12---your r stat has an extra space after the (

Page 16 in the mark-up looks like you have deleted some parts that need to be in—please check

Page 20 you have 4four—you need to have only 1 or the other. Given that mostly you are using numbers the 4 might be the best

Page 25 to many lines between last paragraph and last heading----last heading is also an orphan

Page 26 Trafimov with the red line needs and &

Page 31—men really had a mean of 73.8? I thought tha average was 7

Reviewers' comments:

Reviewer's Responses to Questions

**Comments to the Author**

1. If the authors have adequately addressed your comments raised in a previous round of review and you feel that this manuscript is now acceptable for publication, you may indicate that here to bypass the “Comments to the Author” section, enter your conflict of interest statement in the “Confidential to Editor” section, and submit your "Accept" recommendation.

Reviewer #2: All comments have been addressed

2. Is the manuscript technically sound, and do the data support the conclusions?

Reviewer #2: Yes

3. Has the statistical analysis been performed appropriately and rigorously? 

Reviewer #2: Yes

4. Have the authors made all data underlying the findings in their manuscript fully available?

Reviewer #2: Yes

5. Is the manuscript presented in an intelligible fashion and written in standard English?

Reviewer #2: Yes

6. Review Comments to the Author

Reviewer #2: (No Response)

7. PLOS authors have the option to publish the peer review history of their article (what does this mean?). If published, this will include your full peer review and any attached files.

Reviewer #2: No

---

## [Author Response · Author response to Decision Letter 1]

13 Nov 2024

Response to editor’s comments:

 Page 5 end of first partial paragraph THE median and mean

Added. Thank you. 

Page 5 u (e.g., educational tests; Ho& Yu, 2015) … In your marked up version on page 5 the sentence makes no sense to me there is something wrong there … [sic[

Your comment wasn’t exactly clear to us, but we understand that the topic sentence could have been clearer. We have therefore reorganized and shortened the sentence and merged the paragraph. The end of the paragraph now reads: 

“Questions about how to characterize distributions are particularly important because many distributions in the social sciences are skewed (Ho & Yu, 2015; Heathcote, Popiel, & Mewhort, 1991), and reliance on the mean to describe skewed distributions can lead to misinterpretations (Balota & Yap, 2011; Micceri, 1989; Wiedermann et al., 2022).”

Seems like your Galton quote needs a page number.

Galton’s 1907 Nature article is less than two pages long. 

(https://galton.org/essays/1900-1911/galton-1907-vox-populi.pdf)

End of page 6 The present study is an orphan header and need two lines of text below it

Same with Method at the end of the next page

We have moved the header to the next page.

In study one why did you use an ANOVA when you knew the distribution was skewed. Also, with numbers this large Field states that most of them have already approached normal —

ANOVAs are statistical tests based on the sampling distribution of the mean. Thanks to the central limit theorem, the sampling distribution of the mean will be normally distributed even with skewed population distributions.

Page 12---your r stat has an extra space after the (

Fixed. Thank you.

Page 16 in the mark-up looks like you have deleted some parts that need to be in—please check

We have checked this section, and it seems as if we have removed only the section that we intended. We agree that the marked-up version was hard to follow in this section, but the regular version was satisfactory..

Page 20 you have 4four—you need to have only 1 or the other. Given that mostly you are using numbers the 4 might be the best

The “4four” is an artifact in the marked version. It seems that it should be written out as “four.” Per APA 7 guidelines: Section 6.32, use numerals to express numbers 10 or above (e.g., 11, 23, 256); Section 6.33, write out numbers as words to express numbers up to nine (e.g., three, seven, eight). There are some exceptions (numbers that represent time, etc.). 

Page 25 to many lines between last paragraph and last heading----last heading is also an orphan

Fixed. Thank you

Page 26 Trafimov with the red line needs and &

Fixed. Thank you

Page 31—men really had a mean of 73.8? I thought tha average was 7

The average for men across all three studies was 73.8, but the mode was still 1.0.

---

## [Editor Report · Decision Letter 2]

19 Nov 2024

PONE-D-23-30826R2(Mis)estimation of the modal number of desired sexual partnersPLOS ONE

Dear Dr. Shapiro,

Thank you for submitting your manuscript to PLOS ONE. After careful consideration, we feel that it has merit but does not fully meet PLOS ONE’s publication criteria as it currently stands. Therefore, we invite you to submit a revised version of the manuscript that addresses the points raised during the review process.

We look forward to receiving your revised manuscript.

Kind regards,

Mary Diane Clark, PhD

Academic Editor

PLOS ONE

Journal Requirements:

Additional Editor Comments:

Here you go--if you simply change the discussion it will be a great paper.

Besr Diane

---

## [Author Response · Author response to Decision Letter 2]

20 Nov 2024

In response to your comments, we have modified the General Conclusion section of the manuscript so that it more directly calls attention to our findings.

---

## [Editor Report · Decision Letter 3]

25 Nov 2024

(Mis)estimation of the modal number of desired sexual partners

PONE-D-23-30826R3

Dear Dr. Shapiro,

We’re pleased to inform you that your manuscript has been judged scientifically suitable for publication and will be formally accepted for publication once it meets all outstanding technical requirements.

Kind regards,

Mary Diane Clark, PhD

Academic Editor

PLOS ONE

Additional Editor Comments (optional):

Thank you so much for working with us on making the suggested changes. The manuscript is now clearer and contributes important information to the literature.
---

## [Editor Report · Acceptance letter]

9 Dec 2024

PONE-D-23-30826R3 

PLOS ONE

Dear Dr. Shapiro, 

I'm pleased to inform you that your manuscript has been deemed suitable for publication in PLOS ONE. Congratulations! Your manuscript is now being handed over to our production team.

Kind regards, 

on behalf of

Dr. Mary Diane Clark 

Academic Editor

PLOS ONE